# Advances in Plant Lipid Metabolism Responses to Phosphate Scarcity

**DOI:** 10.3390/plants11172238

**Published:** 2022-08-29

**Authors:** Shengnan Zhu, Cuiyue Liang, Jiang Tian, Yingbin Xue

**Affiliations:** 1Life Science and Technology School, Lingnan Normal University, Zhanjiang 524048, China; 2Root Biology Center, State Key Laboratory for Conservation and Utilization of Subtropical Agro-Bioresources, College of Natural Resources and Environment, South China Agricultural University, Guangzhou 510642, China; 3College of Coastal Agricultural Science, Guangdong Ocean University, Zhanjiang 524088, China

**Keywords:** lipid metabolism, phosphate starvation, triacylglycerol, phospholipid

## Abstract

Low phosphate (Pi) availability in soils severely limits crop growth and production. Plants have evolved to have numerous physiological and molecular adaptive mechanisms to cope with Pi starvation. The release of Pi from membrane phospholipids is considered to improve plant phosphorus (P) utilization efficiency in response to Pi starvation and accompanies membrane lipid remodeling. In this review, we summarize recent discoveries related to this topic and the molecular basis of membrane phospholipid alteration and triacylglycerol metabolism in response to Pi depletion in plants at different subcellular levels. These findings will help to further elucidate the molecular mechanisms underlying plant adaptation to Pi starvation and thus help to develop crop cultivars with high P utilization efficiency.

## 1. Introduction

Phosphorus (P), as an essential macronutrient for plants, is a vital component of nucleic acids, membrane lipids, and adenosine triphosphate (ATP), and is involved in various metabolic processes [1,2]. The phosphate (Pi) directly absorbed by plants is easily fixed in soil, which leads to low P availability, which then severely limits crop growth and production [3]. A traditional method used to manage P deficiency is to increase the application of Pi fertilizers, which leads to the excessive waste of Pi rock resources and environmental eutrophication [4,5]. Over the past few decades, the underlying molecular mechanisms and genetic bases of strategies that plants use to adapt to P deficiency have been extensively investigated; these strategies include changes in root morphology or architecture, the promotion of the expressions of Pi transporter genes, the enhancement of the excretions of organic acids and acid phosphatases, and so on [6,7]. In addition to improving the efficiency of Pi acquisition from soil, an increasing number of studies have also focused on plants’ internal P pools for metabolism and remobilization, such as phospholipids [8,9]. Phospholipids store ~30% of cellular organic P, which is an organic P pool; it is used as an internal Pi source that enables plants to cope with Pi depletion [4,10].

Lipid metabolism, especially in terms of membrane lipid remodeling, has been shown to be an indispensable and well-conserved adaptive mechanism for plants’ adaption to Pi deprivation [8,9,11]. Most extraplastidic membranes are mainly composed of phospholipids such as phosphatidic acid (PA), phosphatidylcholine (PC), phosphatidylethanolamine (PE), phosphatidylinositol (PI), and phosphatidylserine (PS) [11,12]. Under Pi-starvation stress, membrane phospholipids are hydrolyzed to release Pi groups. This process mainly involves three pathways: the phospholipase C (PLC) pathway, the phospholipase D (PLD) and phosphatidic acid phosphatase (PAP) pathway, and the lipid acyl hydrolase (LAH) and glycerophosphodiester phosphodiesterase (GDPD) pathway [8]. Meanwhile, to maintain membrane functionality, non-P galactolipids including monogalactosyldiacylglycerol (MGDG) and digalactosyldiacylglycerol (DGDG) or sulfolipids (i.e., sulfoquinovosyldiacylglycerol, SQDG) are synthesized to replace phospholipids and to be incorporated into the membrane [8,11]. It is worth noting that the synthesis of non-P galactolipids and sulfolipids mainly takes place in plastids, in which membranes mainly include MGDG, DGDG, SQDG, and phospholipid phosphatidylglycerol (PG) [8,12,13]. Therefore, membrane lipid remodeling involves two main processes: the hydrolysis of phospholipids and the synthesis of glycolipids. Under low-P stress conditions, sets of key genes or proteins that participate in the decreases in phospholipids and increases in non-phosphorus lipids in membrane lipid remodeling have been reported in several species, such as Arabidopsis (*Arabidopsis thaliana*), rice (*Oryza sativa*), oat (*Avena sativa*), tomato (*Solanum lycopersicum*), and soybean (*Glycine max*) [10,12,14,15,16]. In this review, we summarize the recent advancements in plants’ lipid metabolism responses to P deficiency and the key genes involved in these pathways. This review not only provides perspectives on the molecular mechanisms underlying plant adaptations to Pi deficiency, but also offers strategies that could be used in breeding crop cultivars with phospholipid-based plant P-use efficiency (PUE).

## 2. Phospholipids’ Degradation in Response to P Deficiency

### 2.1. Phospholipase C Pathway

The plant phospholipase C family includes the phosphoinositide (PI)-specific phospholipase C (PI–PLC) and non-specific phospholipase C (NPC) families according to their substrate specificity [17,18]. Accumulating evidence has shown that PI-PLCs participate in the hydrolyzation of phosphatidylinositol 4,5-bisphosphate to generate inositol 1,4,5-trisphosphate and DAG [19]. The plasma membrane’s perception of external stimuli may be changed and triggered to activate lipid signaling to regulate diverse cellular processes [18]. For example, inositol 1,4,5-trisphosphate, one product of phosphatidylinositol 4,5-bisphosphate hydrolyzed by PI-PLCs, is involved in Ca^2+^ release [18]. There are nine PI-PLC members in Arabidopsis named AtPLC1 to AtPLC9; most of them are induced by diverse environment stresses [19,20,21]. Among them, AtPLC2 has been identified as a primary PI-PLC and plays a role in the phospholipid metabolism and endoplasmic reticulum stress responses [22]. However, none of these Arabidopsis *PI-PLC* genes have been reported to be involved in responses to P deficiency.

NPCs, which are different than PI-PLCs and belong to the non-specific phospholipase C family, can function in diverse phospholipids, including phosphatidylcholine (PC), phosphatidylethanolamine (PE), and phosphatidylserine (PS) [8,17]. A total of six members of the Arabidopsis NPC family have been identified, namely AtNPC1–AtNPC6 [23]. Among them, the transcript levels of two members are induced by P deficiency (Table 1). AtNPC4 has been reported to be the primary contributor to NPC activity under Pi-starvation conditions [23]. Recently, AtNPC4 was confirmed to mainly function in the lipid metabolism in roots during short-term Pi depletion by hydrolyzing PC, the most abundant glycerophospholipid in vitro [23,24,25]. However, the knockout of *AtNPC4* had no effects on the level of PC, suggesting that AtNPC4 is not the predominant enzyme of PC [23]. Recent research also showed that AtNPC4 had high hydrolyzing activity on glycosylinositolphosphorylceramide (GIPC), the most abundant phosphosphingolipid in plants, to generate hydroxyceramide (hCer) in vitro [26]. Additionally, the knockout of *AtNPC4* significantly decreased the loss of GIPC under P-deficiency conditions, indicating that AtNPC4 plays a critical role in the degradation of plasma membrane GIPC, but not PC, in response to Pi starvation [26]. Recently, the C terminus acylation of AtNCP4 was reported to be essential to its plasma membrane association and function in response to P deficiency [27]. On the other hand, AtNPC4 mainly functions in the plasma membrane [23], suggesting that AtNPC4 is also involved in lipid-signaling regulation. DAG, as a product of phospholipids hydrolyzed by NPC, was proposed to mediate lipid signaling in the regulation of root development [25,28]. In Arabidopsis, the *atnpc4* mutant showed a shorter root hair length compared to normal plants under Pi-starvation conditions, suggesting that *AtNPC4* plays a role in the plant lipid-signaling response to early stages of P deficiency, mediating root hair elongation [24]. In addition to AtNPC4, AtNPC5 was also induced by Pi starvation, and the subcellular localization of AtNPC5 was mainly in the soluble fraction [25,28]. Furthermore, the *atnpc5* mutant significantly reduced the level of DGDG compared to wild types in Arabidopsis [28], suggesting that *AtNPC5* also plays a role in membrane lipid remodeling.

### 2.2. Phospholipase D and Phosphatidic Acid Phosphatase Pathway

Phospholipase D is also found to participate in the hydrolyzation of phospholipids to generate phosphatidic acid and soluble head groups; then, phosphatidic acid could be further hydrolyzed by phosphatidic acid phosphatase to generate DAG and release Pi [8,46,47]. The Arabidopsis phospholipase D gene family member *AtPLDZ2* was reported to be induced by P deficiency in both shoots and roots [29]. AtPLDZ2 localized in the tonoplast and the relative amounts of PC and PE were significantly increased in the *atpldz2* mutant compared to the WT of Arabidopsis roots under Pi-starvation conditions [29,48], suggesting that *AtPLDZ2* plays a role in tonoplast membrane phospholipids’ degradation in response to Pi starvation (Table 1). Furthermore, the *atpldz2* mutant also decreased the relative amounts of DGDG in Pi-limited Arabidopsis roots, suggesting that *AtPLDZ2* also plays a role in membrane remodeling in roots, especially in DGDG synthesis [29]. AtPLDZ2 was also reported to be involved in root hair development in response to Pi starvation [30]. Under P-deficiency conditions, an increased abundance of AtPLDZ2 proteins derives more PA, which can bind to SORTING NEXIN 1 (SNX1) to decrease vacuole endocytosis and the degradation of PIN-FORMED2 (PIN2), leading to an increase in PIN2 accumulation at the plasma membrane, thus promoting root hair growth [30]. In addition, compared to AtNPC4, AtPLDZ2 also was shown to have greater effects on leaves’ lipid remodeling at later stages of Pi depletion [24].

PAPs have been reported to function in P-deficiency-induced membrane lipid remodeling [8]. According to their enzymatic properties, PAPs can be divided into Mg^2+^-dependent or -independent types [49]. Phosphatidic acid phosphohydrolase (PAH) is an important Mg^2+^-dependent PAP [8,50]. In Arabidopsis, two *PAH* genes (*AtPAH1* and *AtPAH2*) were identified and suggested to function redundantly in PA degradation in leaves’ endoplasmic reticulum (ER) [50]. It is still unknown whether *AtPAH1* and *AtPAH2* are induced by Pi starvation. However, a double mutant using *atpah1* and *atpah2* significantly increased the phospholipid level but decreased the DGDG level, impairing shoot and root growth under P-deficiency conditions [11], suggesting that AtPAH1 and AtPAH2 play roles in the membrane lipid metabolism and remodeling. Lipid phosphate phosphatase (LPP) has been known to belong to the Mg^2+^-independent PAP family and functions in the hydrolysis of diverse substrates, including PA, lyso-PA, and diacylglycerol pyrophosphate [8,9,51]. Although nine LPP members exist in Arabidopsis, none of them are induced by P deficiency [8,52].

### 2.3. Lipid Acyl Hydrolase and Glycerophosphodiester Phosphodiesterase Pathway

LAH can hydrolyze phospholipids to generate fatty acids and glycerophosphodiester (GPD), which is further hydrolyzed by glycerophosphodiester phosphodiesterase (GDPD) to generate glycerol-3-phosphate (G3P) and corresponding alcohols (choline, ethanolamine, inositol, glycerol, etc.) (Figure 1) [31]. In Arabidopsis, several LAH (or PLA) genes were identified as up-regulated by P deficiency [53]; however, the functions of proteins that were involved in the phospholipid metabolism’s response to Pi starvation remain largely unknown. The Arabidopsis GDPD subfamily has six members, all of which are up-regulated by Pi starvation except for *AtGDPD4* [31]. As a plastid-localized protein, the *atgdpd1* mutant exhibited impaired growth coupled to the reduction in both glycerol-3-phosphate and Pi contents in shoots and roots under Pi-starvation conditions [31], suggesting that *AtGDPD1* plays a role in Pi release from phospholipids in response to P deficiency (Table 1). In addition to *AtGDPD1*, Pi-starvation-induced *AtGDPD6* also was functionally characterized to play a role in glycerophosphocholine (GPC) hydrolysis in response to P deficiency [32]. In Arabidopsis, the *atgdpd6* mutant also exhibited decreased root growth under Pi-deficient conditions [32]. However, when G3P was supplemented in the medium, the inhibited root phenotype of the *atgdpd6* mutant was rescued, suggesting that *AtGDPD6* also plays a role in the degradation of glycerophosphodiester and sustains root growth under P-deficiency stress [32]. There are 13 OsGDPD members in rice. Among them, *OsGDPD1/2/3/5* were up-regulated in roots after 7 days of Pi starvation and *OsGDPD1/2/34/5/7/10/11* were up-regulated after 15 days of Pi starvation [34]. Recently, *OsGDPD2* was found to be up-regulated by P deficiency in both shoots and roots, and it was suggested to be downstream of OsPHR2 [33]. Additionally, OsGDPD2 exhibited hydrolysis activities against several glycerophosphodiesters, including glycerophosphocholine (GPC), glycerophosphoinositol (GPI), and glycerophosphoethanolamine (GPE), and played a role in membrane remodeling in response to Pi starvation [33]. In addition, a total of six CaGDPD members were identified in chickpea (*Cicer arietinum*), of which one (*CaGDPD1*) was shown to be up-regulated while three (*CaGDPD2/3/4*) were down-regulated after 15 days of Pi starvation in roots [34]. Moreover, *CaGDPD1* was proved to be localized in the ER and other endomembranes and exhibited high enzyme activity in GPC and glycerophosphoethanolamine (GPE) [34], suggesting that *CaGDPD1* may play a role in chickpea roots’ adaption to P deficiency through the hydrolyzation of glycerophosphodiester (Table 1).

## 3. Non-Phosphorus Lipid Biosynthesis

The compensatory increase in levels of non-phosphorus lipids such as galactolipids and sulfolipids is an important strategy for plants’ adaption to P deficiency [54]. It is known that phospholipids, especially with regard to the PC of extraplastidic membranes, are replaced by DGDG in most seed plants, while PG is replaced by SQDG in plastid membranes, which was attributed to the biosynthesis pathway (Figure 1) [8,55]. 

### 3.1. Biosynthesis of Monogalactosyldiacylglycerol and Digalactosyldiacylglycerol

MGDG is synthesized from DAG by MGDG synthase; subsequently, DGDG can be further synthesized from MGDG by DGDG synthase [8]. Three MGDG synthases (AtMGD1, AtMGD2, and AtMGD3) have been identified in Arabidopsis. According to their N-termina sequence, the three MGDs can be divided into two types [35]. *AtMGD1* belongs to type A and is mainly expressed in green tissues and localized in the envelope of chloroplasts [35]. *AtMGD2* and *AtMGD3*, both of which belong to type B, are highly expressed in nongreen tissue and are localized in plastids [35]. *AtMGD2* and *AtMGD3* are induced by P deficiency (Table 1) but *AtMGD1* is not [35,36]. P deficiency could induce the accumulation of DGDG in roots. Interestingly, the loss of the *atmgd3* mutant or *atmgd2 atmgd3* double mutant was shown to cause lower or even fully abolished DGDG content [37], suggesting that *AtMGD2* and *AtMGD3* may mediate the synthesis of two essential components of membrane lipid remodeling (MGDG and DGDG) in Pi-deficient roots in Arabidopsis. In rice, three MGD members (OsMGD1, 2, and 3) were identified, and all of them were shown to be up-regulated in roots through Pi starvation, especially *OsMGD3* [38,56]. The knockout or overexpression of *OsMGD3* significantly inhibited or enhanced P utilization efficiency and lateral root growth in response to Pi starvation [38]. Furthermore, it was demonstrated that *OsMGD3* was directly regulated by the OsPHR2 transcription factor under P-deficiency conditions [38].

There are two DGDG synthases in Arabidopsis, *AtDGD1* and *AtDGD2*, and both of them are induced by P deficiency (Table 1) [39,40,41]. In Arabidopsis, the *atdgd1* mutant exhibited accumulated DGDG in extraplastidic membranes and the impaired growth of Arabidopsis under P-deficiency conditions [14], suggesting that AtDGD1 plays a general role in DGDG biosynthesis. In addition, the *atdgd2* mutant showed a similar phenotype to that of *atmgd2* and *atmgd3* double mutant in fatty acid species of DGDG, indicating its redundant function [37,39]. 

### 3.2. Biosynthesis of Sulfoquinovosyldiacylglycerol

SQGD, as an anionic lipid, is also an important non-phosphorus membrane lipid that is significantly increased by low-P stress. Under P-deficiency conditions, SQDG is synthesized to substitute for PG to maintain Pi homeostasis in Arabidopsis [8]. The P-deficiency-response genes *UDP-glucose pyrophosphorylase 3* (*UGP3*), *UDP-sulfoquinovose synthase* (*SQD1*), and *sulfolipid synthase* (*SQD2*) were reported to participate in the synthesis of SQDG in Arabidopsis [8,42,44,57]. The synthesis of SQGD takes place in two steps: SQD1 catalyzes the assembly of UDP-sulfoquinovose via uridine diphosphate glucose (UDPG) and sulfite, and then AtSQD2 functions in transferring the sulfoquinovose of UDP-sulfoquinovose to DAG to generate SQDG (Figure 1) [44]. Mutant *sqd2* could reduce Arabidopsis growth under Pi-starvation conditions [44], suggesting that SQD2 may be involved in the substitution of SQDG for anionic phospholipids in response to P deficiency. In rice, the homolog of AtSQD1—OsSQD1—also was identified to be located in the chloroplast and play a role in lipid composition changes and Pi homeostasis in response to P deficiency [43].

## 4. Other Aspects of Lipid Metabolism in Response to P Deficiency

### 4.1. Triacylglycerol Metabolism

P deficiency also induces significant changes in triacylglycerol (TAG) metabolism. It was reported that the content of TAG increased more than 13-fold in both Arabidopsis shoots and roots upon P deficiency [58]. Similar increases in TAG in response to P deficiency also were reported in tomato leaves and roots and soybean leaves [12,16]. The accumulation of TAG is known to store carbon in an inactive form, which has been found in nitrogen limitation and leaf senescence in Arabidopsis [59,60]. The underlying mechanism may be related to *diacylglycerol acyl transferase 1* (*DGAT1*), which catalyzes DAG and acyl-CoA to synthesize TAG and is regulated by the ABA insensitive 4 (ABI4) transcription factor [59,61]. However, neither *DGAT1* nor *ABI4* was induced by Pi starvation in Arabidopsis [58], suggesting that another responsive mechanism underlies Pi-starvation-induced TAG accumulation. Recently, a tomato *DGAT* gene (*SlDGAT2*) was reported to be up-regulated in leaves after seven days of P-deficiency treatment [12], which suggested that P deficiency may be induced by TAG accumulation, although its function requires further characterization. On the other hand, potential phosphorylation sites and interacting proteins were found on DGAT1; thus, post-translational modification or interaction with other proteins can be proposed for DGAT involved in TAG accumulation under Pi-starvation conditions.

### 4.2. Glucuronosyldiacylglycerol Metabolism

Glucuronosyldiacylglycerol (GlcADG) is a novel lipid that is essential to the protection of plants against P deficiency [62]. The Pi-starvation-induced accumulation of GlcADG was reported in Arabidopsis, rice, and soybean [16,62]. Lipidomic analysis in Arabidopsis indicated that the composition of DAG in GlcADG and SQDG was nearly identical, and the plant growth and GlcADG accumulation of the *sqd2* mutant were severely impaired compared to the wild-type control, suggesting that the biosynthesis of GlcADG shares the same pathway with SQDG synthesis in chloroplasts [62]. Thus, GlcADG is synthesized by SQD2 using DAG as the substrate and UDP glucuronic acid (UDP-GlcA) as a sugar donor [62]. In soybean leaves, it seems that Pi starvation could not increase the content of SQDG but increased the content of GlcADG, suggesting that GlcADG can be an alternative to accumulated SQDG in response to P limitation in some plant species [16], which enables energy conservation in response to Pi starvation [16].

### 4.3. The De Novo Biosynthesis Pathway of Glycerolipid

The glycerolipid de novo biosynthesis pathway (the Kennedy pathway) includes two steps: G3P acyltransferase (GPAT) converts G3P to lysophosphatidic acid (LPA), then the LPA is further converted to PA by lysophosphatidic acid acyltransferase (LPAT) [8,45,63]. This pathway occurs in both plastids and ER, in which the PA is further used for synthesis of phospholipids, glycolipids, and TAG [63]. This de novo biosynthesis pathway was considered to be essential in membrane lipid supplement to promote root growth under Pi starvation [45,64]. Among the five *LPAT* members of Arabidopsis, *AtLPAT2* was significantly induced by Pi starvation in roots and localized in ER [45]. Overexpression of *AtLPAT2* significantly increased the root length and the amount of PC in Arabidopsis root under Pi starvation [45]. These results suggested that Pi-starvation-induced *AtLPAT2* functions in mediating de novo phospholipid biosynthesis and root development under Pi starvation in Arabidopsis [45].

## 5. Key Regulators in Lipid Metabolism

P-deficiency-responsive membrane lipid remodeling has been reported to be an important adaptive strategy that involves several lipid species’ metabolites and diverse Pi-starvation-responsive genes. However, the key regulators and regulatory mechanisms involved in these responses remain largely unknown. Phosphate starvation response regulator 1 (PHR1) is known as the global regulator of the P-deficiency response [6]. Phospholipids’ metabolism and membrane lipid remodeling in response to Pi starvation are also partially regulated by PHR1 in Arabidopsis. For example, P-deficiency-induced *AtNCP5*, *AtPLDZ2*, *AtMGD2/3*, and *AtDGD1* were partially attenuated and *AtNCP4* was evenly and completely suppressed in *phr1*-mutant Arabidopsis [58,65]. Furthermore, a large number of these Pi-starvation-responsive genes contain one or more P1BS motifs (PHR1 protein binding site) in their promoters, except for *AtMGD3* [58]. The changes in lipid composition in response to P deficiency, such as the decrease in phospholipids and increase in non-phosphorus lipids, were also altered in *phr1*-mutant Arabidopsis [58]. These results indicated that the PHR1 transcription factor plays a role in membrane lipids’ remodeling in response to P-deficiency stress (Figure 2). Still, the application of phosphite, which is not metabolically available for plants, can obstruct the Pi-starvation responses in changes in membrane lipids and several related genes [66], suggesting that Pi-sensing signal transduction plays a key role in Pi-starvation-induced lipid responses rather than the damage caused by Pi starvation. In Arabidopsis, it was reported that Pi-starvation-induced membrane lipid alteration was regulated by Pi signaling and auxin/cytokinin cross-talk [66]. In addition, auxin played a positive regulatory role in activating *AtMGD2/3* expression, and was indispensable in the Pi-starvation response, while cytokinin played a negative regulatory role [66]. Furthermore, auxin-signaling-mediated membrane lipid remodeling during P deficiency was revealed by auxin response factors ARF7/19 and IAA14, repressors of auxin signaling, in Arabidopsis [67]. A low P level enhanced the auxin sensitivity by increasing the expression of *TIR1*; the auxin transport from shoot to root could induce the degradation of IAA14 to release ARF7/19, thus regulating the cell-cycle activities and other responses to mediate gene expressions and membrane lipid remodeling [66,67,68]. However, ARF7 and ARF19 did not seem to directly regulate these Pi-starvation-response genes, including *MGD3* and *IPS1*, in Arabidopsis under P-deficiency conditions [67], indicating that other unknown regulators exist (Figure 2). Jasmonic acid (JA) also was reported to play a role in the membrane remodeling triggered by Pi starvation [69]. In Arabidopsis, JA impacted the level of MGDG by regulating the expression of *AtMGD3* [69]. However, the underlying molecular mechanism remains unknown. Recently, the JASMONATE ZIM-DOMAIN (JAZ) proteins, while functioning as transcriptional repressors of JA signaling, were reported to play a role in connecting JA and Pi signaling [70]. For example, *OsJAZ11*, a Pi-starvation-responsive gene regulated by OsPHR2 regulated root growth and Pi homeostasis by suppressing JA signaling in rice roots [70]. These results suggested that JAZ proteins may play a role in JA-mediated membrane metabolism in response to Pi starvation, which requires further confirmation.

## 6. Conclusions and Perspectives

To cope with P-deficiency stress, diverse strategies have evolved in plants to enhance their external Pi uptake capacities and improve their internal P use efficiencies [71,72,73,74,75]. Under P-deficiency conditions, membrane phospholipids can be hydrolyzed for Pi reutilization; meanwhile, galactolipids and sulfolipids are synthesized to compensate for the degraded phospholipids [10,14]. This membrane lipid remodeling is known to be a critical strategy for plants’ adaption to Pi starvation. This review mainly focused on how P-deficiency stress affects membrane lipid remodeling by P-deficiency-responsive genes involved in this process. It is clearly apparent that several P-deficiency-response genes are present in all three phospholipid degradation pathways, including *NPCs*, *PLDs*, and *GDPDs* (Table 1). Similarly, both the galactolipid and sulfolipid biosynthesis pathways are also induced by P deficiency, as reflected by the up-regulated *MGD*, *DGD*, and *SQD* levels identified in the synthesis of MGDG, DGDG, and SQDG, respectively (Table 1). These candidate genes helped to preliminarily reveal the mechanisms underlying membrane lipids’ metabolism response to Pi starvation and thus contributed to lipid-based crop PUE improvement. However, many P-deficiency-responsive genes remain missing from the scheme. For example, Pi-starvation-induced *PI-PLC* genes that function in phosphoinositide hydrolysis remain unknown, although several Arabidopsis *PI-PLC* family member genes, such as *AtPLCs,* were expressed in response to other environmental stresses [19,20,21]. Similarly, the functions of Pi-starvation-induced, phosphatidic-acid-hydrolysis-related genes and lipid acyl hydrolase genes remain largely unknown. In addition, P deficiency significantly increased the accumulation of TAG in plants’ leaves and roots [12,16,58], and these molecular mechanisms and P-deficiency-responsive genes require further characterization. In *phr1*-mutant Arabidopsis, the Pi-starvation-induced, lipid-metabolism-related genes such as *AtNCP5*, *AtPLDZ2*, *AtMGD2/3*, and *AtDGD1* were partially attenuated [58,65], indicating that other regulators contribute to the lipid metabolism under Pi-starvation conditions. In addition, the expression levels of several Pi-starvation-induced, lipid-remodeling-related genes can be obstructed by phosphite, suggesting that other regulators involved in Pi-sensing transduction, which play key roles in this process, may exist [66]. Thus, the key regulator of P-deficiency-induced lipid remodeling requires further confirmation. On the other hand, most of these accumulative results mainly came from research regarding model Arabidopsis and rice plants; thus, it is important to explore the complex mechanisms underlying membrane lipid remodeling in other plants.

## Figures and Tables

**Figure 1 plants-11-02238-f001:**
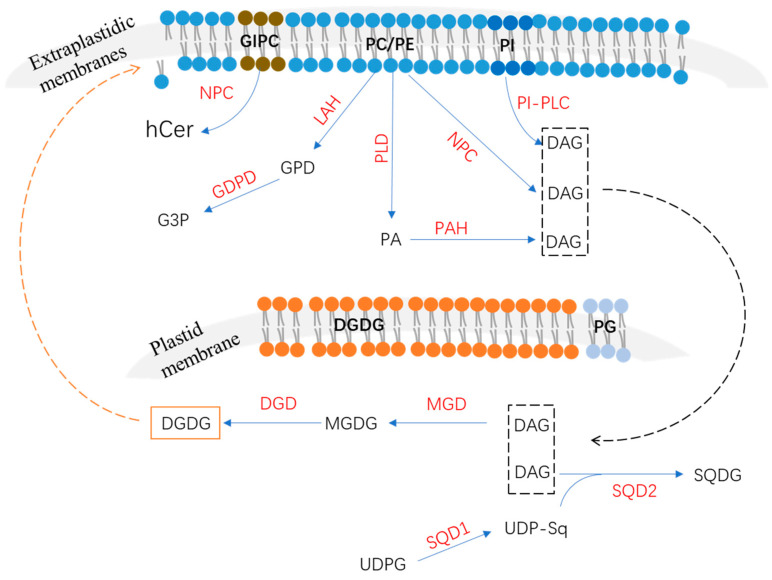
Metabolic pathways in membrane lipid remodeling and related enzymes. The plastid membranes mainly consist of glycolipids, while extraplastidic membranes mainly consist of phospholipids. Under P-deficiency conditions, phospholipids can be degraded to release the phosphate group, then the non-phosphorus galactolipids are compensatively synthesized to replace the phospholipids. GIPC, glycosylinositolphosphorylceramide; hCer, hydroxyceramide; PC, phosphatidylcholine; PE, phosphatidylethanolamine; PG, phosphatidylglycerol; GPD, glycerophosphodiester; G3P, glycerol-3-phosphate; PI, phosphatidylinositol; PA, phosphatidic acid; DAG, diacylglycerol; DGDG, digalactosyldiacylglycerol; MGDG, monogalactosyldiacylglycerol; SQDG, sulfoquinovosyldiacylglycerol; UDPG, uridine diphosphate glucose; UDP-sq, UDP-sulfoquinovose; MGD, MGDG synthase; DGD, DGDG synthase; SQD1, UDP-sulfoquinovose synthase; SQD2, sulfolipid synthase.

**Figure 2 plants-11-02238-f002:**
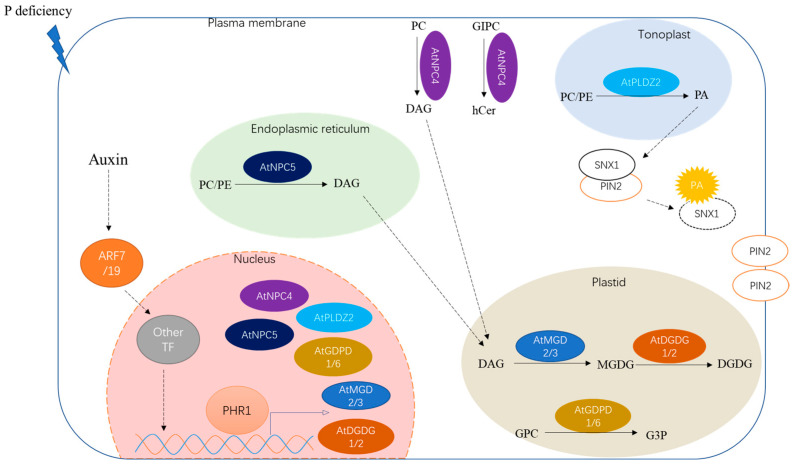
Proposed model for membrane lipid metabolisms in response to P deficiency in Arabidopsis roots. The enzymes induced by P deficiency are shown in different organelles according to their localizations. PHR1 and ARF7/19 are the transcription factors. GIPC, glycosylinositolphosphorylceramide; hCer, hydroxyceramide; PC, phosphatidylcholine; PE, phosphatidylethanolamine; G3P, glycerol-3-phosphate; PA, phosphatidic acid; DAG, diacylglycerol; DGDG, digalactosyldiacylglycerol; MGDG, monogalactosyldiacylglycerol; GPC, glycerophosphocholine; MGD, MGDG synthase; DGD, DGDG synthase; SNX1, SORTING NEXIN 1; PIN2, PIN-FORMED2.

**Table 1 plants-11-02238-t001:** P-deficiency-response genes involved in membrane lipid metabolism.

Gene	Species	Pi Starvation	Tissues	Subcellular Localization	Pathway	Functions in Adaption to Pi Starvation	References
*AtNPC4*	*Arabidopsis thaliana*	Up-regulated	Shoot and root	Plasma membrane	Hydrolyzing PC to generate DAG. Hydrolyzing GIPC to generate hCer.	Knockout of *AtNPC4* decreases the loss in GIPC and impedes root growth under Pi starvation.	[23,24,26,27]
*AtNPC5*	*Arabidopsis thaliana*	Up-regulated	Leaf	Cytosol	Hydrolyzing PC and PE to generate DAG.	Knockout of *AtNPC5* decreases the accumulation of DGDG in both leaf and root under Pi starvation.	[28]
*AtPLDX2*	*Arabidopsis thaliana*	Up-regulated	Shoot and root	Tonoplasts	Hydrolyzing PC and PE to generate PA.	Knockout of *AtPLDZ2* decreases the amount of DGDG and increases the amounts of PC and PE in root, decreases primary root length, and increases root hair density and root hair length under Pi starvation.	[24,29,30]
*AtGDPD1*	*Arabidopsis thaliana*	Up-regulated	Shoot and root	Plastids	Hydrolyzing glycerophosphodiesters into G3P and the corresponding alcohols.	Knockout of *AtGDPD1* decreases G3P content, Pi content, and seedling growth rate under Pi starvation.	[31]
*AtGDPD6*	*Arabidopsis thaliana*	Up-regulated	Flower and primary root	Nd	Hydrolyzing GPC to generate G3P.	Overexpression of *AtGDPD6* increases root length and knockout of *AtGDPD6* decreases root length under Pi starvation.	[32]
*OsGDPD2*	*Oryza* *sativa*	Up-regulated	Shoot and root	Nd	Hydrolyzing glycerophosphodiesters into G3P and the corresponding alcohols.	Overexpression of *OsGDPD2* increases Pi content, root growth, and biomass accumulation under Pi starvation.	[33]
*CaGDPD1*	*Cicer arietinum*	Up-regulated	Root	Endoplasmic reticulum	Having enzyme activities on GPC and GPE.	Nd.	[34]
*AtMGD2*	*Arabidopsis thaliana*	Up-regulated	Root	Plastid	Catalyzing DAG and UDP-Gal into MGDG.	Knockout of *AtMGD2* decreases root length under Pi starvation.	[35,36,37]
*AtMGD3*	*Arabidopsis thaliana*	Up-regulated	Root	Plastid	Catalyzing DAG and UDP-Gal into MGDG.	Knockout of *AtMGD3* decreases root length and shoot and root fresh weight under Pi starvation.	[35,36,37]
*OsMGD3*	*Oryza sativa*	Up-regulated	Root	Plastid	Catalyzing DAG and UDP-Gal into MGDG.	Knockout of *OsMGD3* decreases shoot dry weight and Pi use efficiency; overexpression of *OsMGD3* increases shoot dry weight, lateral root number, root Pi acquisition efficiency, and total P content per plant shoot under Pi starvation.	[38]
*AtDGD1*	*Arabidopsis thaliana*	Up-regulated	Leaf	Chloroplast	Catalyzing MGDG and UDP-Gal into DGDG.	Knockout of *AtDGD1* decreases the amount of DGDG under Pi starvation.	[14,39]
*AtDGD2*	*Arabidopsis thaliana*	Up-regulated	Leaf	Chloroplast	Catalyzing MGDG and UDP-Gal into DGDG.	Knockout of *AtDGD2* decreases the amount of DGDG under Pi starvation.	[14,39,40,41]
*AtSQD1*	*Arabidopsis thaliana*	Up-regulated	Leaf	Chloroplast	Catalyzing UDPG and sulfite into UDP-sulfoquinovose.	Nd.	[42]
*OsSQD1*	*Oryza sativa*	Up-regulated	Shoot and root	Chloroplast	Catalyzing UDPG and sulfite into UDP-sulfoquinovose.	Knockout of *OsSQD1* decreases root growth and increases the Pi and total P concentration of shoot and root under Pi starvation.	[43]
*AtSQD2*	*Arabidopsis thaliana*	Up-regulated	Leaf	Chloroplast	Catalyzing UDP-sulfoquinovose and DAG into SQDG; catalyzing UDP-GlcA and DAG into GlcADG.	Knockout of *AtSQD2* decreases the amount of SQDG and reduces plant fresh weight under Pi starvation.	[44]
*SlDGAT2*	*Solanum lycopersicum*	Up-regulated	Leaf and root	Nd	Catalyzing DAG and acyl-CoA into TAG.	Nd.	[12]
*AtLPAT2*	*Arabidopsis thaliana*	Up-regulated	Root	Endoplasmic reticulum	Catalyzing LPA into PA.	Overexpression of *AtLPAT2* increases the length of primary root and the amount of PC in root under Pi starvation.	[45]

Nd., not described in the studies.

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
