# Peer review of "Advances in Plant Lipid Metabolism Responses to Phosphate Scarcity"

_plants, 2022, doi:10.3390/plants11172238_

Round 1
Reviewer 1 Report
This review summarizes and organizes the genes and pathways involved in plants' coping mechanisms with low-phosphorus conditions. It would be a helpful reference for the field to quickly locate relevant primary studies.
Several minor comments:
1) The authors should be consistent with their gene/protein nomenclature (whether to capitalize or italicize). For example, "atnpc4 mutant" in line 102 and "AtNPC4" in line 103.
2) There are numerous typos and grammar mistakes
3) "key/imporant" role are overused and deflated the importance.
Author Response
1)The authors should be consistent with their gene/protein nomenclature (whether to capitalize or italicize). For example, "atnpc4mutant" in line 102 and "AtNPC4" in line 103.
Answer:Thank you for your helpful comments. The gene names are capitalized and italicized, the mutant names are lowercase and italicized, and the protein names are capitalized and not italicized. We have checked all the names in our manuscript.
2)There are numerous typos and grammar mistakes
Answer:Thank you for your helpful comments. We have further checked our manuscript and used English editing service to check the grammar.
3) "key/imporant" role are overused and deflated the importance.
Answer:Thank you for your helpful comments. We have deleted or changed the description for several “key/important” in our revised manuscript.
Reviewer 2 Report
Dear, Yingbin Xue, Ph.D.
Please check and revise.
-------------------------------------------------------------------------------------------------------------------------
I thought there was no revision.
-------------------------------------------------------------------------------------------------------------------------
Author Response
Thank you for your helpful comments. We have further checked and revised our manuscript.
Reviewer 3 Report
The manuscript entitled " Advances in Plant Lipid Metabolism Responses to Phosphate Scarcity" is well written review article with up to date information on the topic. The article is worth publishing and will benefit the researcher in the field.
I have following suggestions to improve the article
1. Recently, it is reported that LPAT2 is a key gene that play a major role in phospholipid biosynthesis and root elongation under phosphate starvation (Angkawijaya et al., 2017, Plant Cell & environment). I suggest to include this information in the article.
2. Besides Auxin and Cytokinin, Jasmonic acid also plays a major role during pi homeostasis in roots (Bipin K pandey et al.,2021 Planta). I suggest to include this information in the article.
3. The ultimate goal of understanding the molecular mechanism underlying lipid remodeling during phosphate starvation is to develop the crops to resist phosphate starvation. I recommend including a table that represent over expressed or knockdown genes in plants to cope with phosphate starvation.
Author Response
1. Recently, it is reported that LPAT2 is a key gene that play a major role in phospholipid biosynthesis and root elongation under phosphate starvation (Angkawijaya et al., 2017, Plant Cell & environment). I suggest to include this information in the article.
Answer:Thank you for your helpful comments. We have included this information as “Jasmonic acid (JA) has also been reported to play a role in the membrane remodeling triggered by Pi starvation [69]. In Arabidopsis, JA can impact the level of MGDG through regulating the expression of AtMGD3 [69]. However, the underlying molecular mechanism remains unknown. Recently, the JASMONATE ZIM-DOMAIN (JAZ) proteins, functioning as transcriptional repressors of JA signaling, is reported to function in connecting JA and Pi signaling [70]. For example, OsJAZ11, a Pi starvation responsive gene regulated by OsPHR2, can regulate root growth and Pi homeostasis by suppressing JA signaling in rice roots [70]. These suggest that JAZ proteins may play a role in JA mediated membrane metabolism in response to Pi starvation, which needs to be further confirmed.” Please check this in details in the revised manuscript on line 313 to line 322.
2. Besides Auxin and Cytokinin, Jasmonic acid also plays a major role during pi homeostasis in roots (Bipin K pandey et al.,2021 Planta). I suggest to include this information in the article.
Answer:Thank you for your helpful comments. We have included this information as “ 4.3. The de novo biosynthesis pathway of glycerolipid
The glycerolipid de novo biosynthesis pathway (the Kennedy pathway) includes two steps: the G-3-P acyltransferase (GPAT) converts G-3-P to lysophosphatidic acid (LPA), and the LPA is further converted to PA by lysophosphatidic acid acyltransferase (LPAT) [8,62,63]. This pathway occurs in both plastids and ER, in where, the PA is further used for synthesis of phospholipids, glycolipids, and TAG [63]. This de novo biosynthesis pathway has been considered to be essential in membrane lipids supplement to promote root growth under Pi starvation [62,64]. Among Arabidopsis five LPAT members, AtLPAT2 is significantly induced by Pi starvation in roots and localized in ER [62]. Overexpression of AtLPAT2 significantly increases the root length and the amount of PC in Arabidopsis root under Pi starvation [62]. These suggest that Pi-starvation induced AtLPAT2 functions in mediating de novo phospholipid biosynthesis and root development under Pi starvation in Arabidopsis [62].” Please check this in details in the revised manuscript on line 269 to line 281.
3. The ultimate goal of understanding the molecular mechanism underlying lipid remodeling during phosphate starvation is to develop the crops to resist phosphate starvation. I recommend including a table that represent over expressed or knockdown genes in plants to cope with phosphate starvation.
Answer:Thank you for your helpful comments. We have added these information in Table 1 on the item of “Functions in adaption to Pi starvation”, in which we listed the phenotypes of overexpression or knockout the genes under Pi starvation.
Reviewer 4 Report
The submitted manuscript entitled “Advances in plant lipid metabolism responses to phosphate scarcity“ presents a comprehensive review of studies on discoveries and the molecular basis of membrane phospholipid alteration and triacylglycerol metabolism in response to Pi depletion in plants. The presented summary and interpretation of the recent researchers can be considered helpful for further elucidation of molecular mechanisms underlying the problem of plant adaptation to Pi starvation. Thus, the presented manuscript can be considered an interesting summary for the scientist engaged in this area of research. The review is detailed, logically structured, and comprehensive. I have read the work thoroughly and cannot point out any substantial allegations against the research design. I would recommend it for publication in the presented form.
However, some linguistic mistakes were found by the reviewer, and as the reviewer is not able to correct the text, it can be predicted that some more could be found in the text:
Line 26 “will severely limit” not “limits”
Line 84 primary not primarily contributor
Line 123 “is also reported involved” – some worlds are missing in this sentences
Line 153 should be “ the reduction” not “the reduce”
Line 178 compensatory not compensatorily
Line 191 “P deficiency cold induce” – the sentence seems incorrect, it should be “induces”
Line 269 metabolites not metabolistes
Line 326 environmental not environment
Generally, articles "the" and "a" seems to be missing in the entire text
Author Response
Thank you for your helpful comments. We have further checked our manuscript and used English editing service to improve it. Please check this in details in the revised manuscript.